# Finite Element Analysis of Normal and Dysplastic Hip Joints in Children

**DOI:** 10.3390/jpm13111593

**Published:** 2023-11-10

**Authors:** Zsuzsánna Incze-Bartha, Sandor Incze-Bartha, Zsuzsánna Simon Szabó, Andrei Marian Feier, Vlad Vunvulea, Ioan Alin Nechifor-Boilă, Ylenia Pastorello, Dezso Szasz, Lóránd Dénes

**Affiliations:** 1Department of Anatomy, “George Emil Palade” University of Medicine, Pharmacy, Science and Technology of Targu Mures, 540139 Targu Mures, Romania; zsuzsanna.incze-bartha@umfst.ro (Z.I.-B.);; 2Department of Orthopedics and Traumatology, “Fogolyan Kristof” County Hospital Sfantu Gheorghe, 520064 Covasna, Romania; 3Department of Orthopaedics and Traumatology, “George Emil Palade” University of Medicine, Pharmacy, Science, and Technology of Targu Mures, 540142 Targu Mures, Romania

**Keywords:** developmental hip dysplasia, children, finite element, geometric model, hip joint, pediatric hip joint

## Abstract

From a surgical point of view, quantification cannot always be achieved in the developmental deformity in hip joints, but finite element analysis can be a helpful tool to compare normal joint architecture with a dysplastic counterpart. CT scans from the normal right hip of an 8-year-old boy and the dysplastic left hip of a 12-year-old girl were used to construct our geometric models. In a three-dimensional model construction, distinctions were made between the cortical bone, trabecular bone, cartilage, and contact nonlinearities of the hip joint. The mathematical model incorporated the consideration of the linear elastic and isotropic properties of bony tissue in children, separately for the cortical bone, trabecular bone, and articular cartilage. Hexahedral elements were used in Autodesk Inventor software version 2022 (“Ren”) for finite element analysis of the two hips in the boundary conditions of the single-leg stance. In the normal hip joint on the cartilaginous surfaces of the acetabulum, we found a kidney-shaped stress distribution in a 471,672 mm^2^ area. The measured contact pressure values were between 3.0 and 4.3 MPa. In the dysplastic pediatric hip joint on a patch of 205,272 mm^2^ contact area, the contact pressure values reached 8.5 MPa. Furthermore, the acetabulum/femur head volume ratio was 20% higher in the dysplastic hip joint. We believe that the knowledge gained from the normal and dysplastic pediatric hip joints can be used to develop surgical treatment methods and quantify and compare the efficiency of different surgical treatments used in children with hip dysplasia.

## 1. Introduction

With an incidence rate of 0.1% among live births, developmental dysplasia stands as the most prevalent orthopedic disorder in neonates. The definition of developmental hip dysplasia encompasses a broad spectrum of atypical hip joint anatomical configurations, ranging from mild dysplasia to subluxation and even complete dislocation of the femoral head from the acetabulum. Over time, these alterations can result in cartilage degeneration and the subsequent onset of secondary osteoarthritis affecting the hip joint. The most commonly encountered deformity in this context is acetabular dysplasia, which compromises the normal architecture of the hip joint, leading to a reduced contact area between articular surfaces and an increase in joint contact pressure [1,2,3,4].

With the implementation of screening procedures, the occurrence of late-discovered cases characterized by ossified physes has become less frequent. Nevertheless, these cases present a formidable challenge in terms of treatment. Various surgical techniques have been outlined to enhance the coverage of the femoral head. Open reduction and hip osteotomies are performed to achieve anatomical correction of the joint. To select the most suitable treatment procedure, it is imperative to grasp the mechanical alterations present in the dysplastic hip joint as compared to its anatomically normal counterpart.

Finite element analysis (FEA) is a valuable tool in the study of normal and dysplastic hip joints in children [5]. FEA allows for the investigation of the mechanical effects of various surgical interventions on the hip joint, as well as the correlation between radiological measurements and mechanical stress [5]. It has been widely applied in biomechanical analyses of hip joints and has shown good agreement with experimental results [6].

In the case of developmental dysplasia of the hip (DDH), FEA has been used to evaluate the effects of rotational acetabular osteotomy (RAO) on the mechanical stress within the hip joint. In a study by Ike et al., subject-specific finite element models were constructed from CT data to analyze the mechanical effects of RAO on the hip joint. The study found that RAO resulted in a redistribution of mechanical stress within the hip joint, which may have implications for the long-term outcomes of the surgery [5]. FEA has also been used to examine the effects of femoroacetabular impingement (FAI) on hip joint mechanical loading.

In a study by Ng et al., subject-specific geometries, kinematics, and kinetics were incorporated into the FEA to better characterize the mechanical stimuli magnitudes and regions. The study hypothesized that hips with cam FAI would demonstrate higher levels of mechanical stress than healthy control hips. The results of the study supported this hypothesis, highlighting the importance of FEA in understanding the mechanical effects of FAI on the hip joint [7].

Furthermore, FEA has been utilized in the analysis of hip joint contact stress during a gait cycle. Inverse dynamic analysis was performed to obtain the dynamic changes of the main hip–femoral muscle force during a gait cycle. These results were then used as boundary and load settings for the hip joint finite element analysis. The study found that hip joint contact stress varied throughout the gait cycle, providing valuable insights into the biomechanics of the hip joint during walking [8].

In addition to its applications in understanding the mechanical behavior of the hip joint, FEA has also been used in the development and evaluation of hip prostheses. The use of FEA has been highlighted in simulating the hip prosthesis under variable loads experienced during normal human activities. This enables designers to develop more reliable hip prostheses that can withstand the demands of daily life [9].

Thus, finite element analysis plays a crucial role in the study of normal and dysplastic hip joints in children. It allows for the investigation of the mechanical effects of surgical interventions, the evaluation of hip joint mechanical loading, and the development of more reliable hip prostheses. By providing valuable insights into the biomechanics of the hip joint, FEA contributes to the understanding and improvement of treatments for hip joint conditions in children.

The initial computational FEA studies examining periprosthetic mechanical alterations were introduced in the 1970s, and this trend persisted into the 1980s. However, a few studies began to delve into research on bone diseases during this period, which saw further development in the 1990s. Notably, the mechanical properties of adult bones were extensively investigated and documented. In the 2000s and 2010s, FEA studies pertaining to the pediatric skeleton emerged, primarily building upon the foundation of adult research [10].

With mathematical model construction and finite element analysis of the reconstructed complex structure, immediate results can be obtained from the behavior of the analyzed structure.

The cartilage contact area and stress distribution across the articular components can be calculated. The aim of this study was to compare the biomechanical characteristics between a normal hip and a dysplastic hip in children [6,7,8,9].

## 2. Materials and Methods

Computed tomography-based finite element models are accurate enough to be as close to biological conditions as possible. Patient-specific volumetric data from computed tomography scans were used to build the geometrical models. The CT data from two patients were processed: a normal right hip from an 8-year-old boy (weighing 24 kg), who presented to our clinic following a motor vehicle accident, and a left hip from a 12-year-old girl (weighing 32 kg) with Tönnis Grade 3 hip dysplasia, who presented to our clinic for preoperative evaluation and surgery.

All examinations were conducted using the Siemens Somatom Sensation, a 16-slice unit manufactured by Siemens Medical Solutions in Forchheim, Germany, following a standardized imaging protocol. The CT scan was performed with the following parameters: a tube voltage of 120 kilovolts (kV), a tube current-time product of 200 milliamperes–seconds (mAs), a pitch value of 0.8, a collimation size of 128 × 0.6 millimeters (mm), and a slice thickness of 3 mm.

Because no other models of child hip geometry could be found, the geometric model of a healthy adult hip was employed, and adult mechanical properties were assigned to the solid bodies. This model was validated against the existing literature models. It is worth noting that the morphology of adult and pediatric skeletons differs significantly, encompassing differences in shape, axis, and long bone rotation. One notable distinction between the material properties of adult and child bones lies in factors such as Young’s modulus, density, yield stress (which tend to be lower in children), and a higher compressive ultimate strain compared with adult bone. Consequently, the structural characteristics of the adult hip differ substantially from those of the pediatric hip, which led to the comparison of a child’s dysplastic hip with a child’s healthy hip.

For the normal hip, 37 slides were processed using a resolution of 0.601563 × 0.601563 mm/pixel in the XY plane with a slice thickness of 3 mm and in the Z-plane distinguishing 256 shades of gray (8-bit). For the dysplastic hip, 28 slides were processed using a resolution of 0.613281 × 0.613281 mm/pixel in the XY plane with a slice thickness of 5 mm and in 8 bits.

### 2.1. The Geometric Model

Using MicroDicom, the data from the CT slices were processed and converted from DICOM format to JPEG [11] and then imported into ImageJ.

The outline of the bony tissue (cortical and trabecular) in the images was traced using the Canny edge detector algorithm. The 8-bit image was simplified into a binary image, mapping the entry points to the surface construction. Most of the boundaries were generated automatically but some of them needed manual correction. The contour points were approximated using NURBS curves, and the irregularity of the contours was smoothed using our own algorithm. In this manner, the number of control points was successfully reduced with minimal distortion [12,13].

The established outlines were over posed using our own software. The surface areas were created using Autodesk Inventor software version 2022. In the tridimensional model construction, we distinguished between the cortical bone, trabecular bone, cartilage, and contact nonlinearities of the hip joint [12,13].

To create the cartilage surface in the joint, each cortical bone was elevated: the acetabular and femoral head with half of the distance between them (2 mm). The visual representation of this process is shown in Figure 1.

In the examined children, only the affected hip was edited. To recreate a complete pelvis for finite element analysis, the contralateral hemipelvis and sacrum were replaced with an edited iliac–pubic bone slab connecting the two articular surfaces. The mathematical model assumed linear elastic and isotropic properties for the bony slab connecting the iliac and pubic bones.

In the final geometrical model construction of the hip joints, all the forming components remained in their original positions imported from the CT scans.

### 2.2. The Mathematical Model

In our models, linear elastic and isotropic properties were assumed for the materials used. During model construction, the material properties were considered separately for the cortical bone, trabecular bone, and articular cartilage. The interfaces between the cortical and trabecular bone, as well as between the trabecular bone and articular cartilage, were fixed. The interface between the cartilaginous surfaces was treated as a nonlinear, frictionless, three-dimensional contact problem.

In the finite element analysis, the density, Young’s modulus, Poisson’s ratio, yield strain, yield stress, ultimate compressive strength, and ultimate tensile strength of the bone mechanical properties of a child were considered, and were represented in Table 1. The primary distinction between adult and child bone characteristics lies in Young’s modulus, density, yield stress (which are lower in children), and a higher compressive ultimate strain compared to adult bone [14,15,16].

For model validation, the mechanical properties of the adult bone were considered [17,18,19].

Autodesk Inventor software version 2022 was used to conduct finite element analysis on the two hips. In the process of generating meshes for the cortical, trabecular bone, and cartilage, we employed linear solid hexahedral elements, as depicted in Figure 2 [20,21].

In the normal hip model, 29,907 hexahedral elements and 55,627 nodes were used. In the dysplastic hip, 27,594 hexahedral elements and 51,436 nodes were used. The numbers of elements and nodes used for different anatomical structures are described in Table 2.

### 2.3. Boundary Conditions

The boundary conditions for the single-leg stance were created in the analysis. A mount of vertical load was placed on the superior aspect of the iliac–pubic bone bridge. The load was calculated using the weights of the patients: 240 N for the normal hip model and 320 N for the dysplastic hip model. Over the middle gluteal muscle attachment, a simulated abductor force was applied. The iliac–pubic bone bridge was fixed on the transverse plane. The distal end of the left femur was fixed in all directions. A nonlinear, frictionless contact was maintained between the cartilaginous surfaces of the hip joints [22,23,24,25].

## 3. Results

The distributions and values of von Mises stress exhibited different patterns in the two analyzed hips. In the normal hip, a band of stress distribution flowed from posterior to anterior and inferiorly through the hemipelvis, originating at the sacroiliac joint and extending through the femoral head to the Adams arch on the femur. The measured values ranged between 2.9 and 8.5 Mpa. The highest stress values were observed at the point of contact between the bony bridge and the iliac joint, as well as the pubic bone, reaching 8.5 Mpa. On the posterior surface of the femoral neck, an area with elevated stress values was identified, which curved anteriorly and extended downward toward the lesser trochanter, as illustrated in Figure 3.

On the cartilaginous surfaces of the acetabulum, a kidney-shaped stress distribution was observed. The measured values ranged from 3.0 Mpa in the anterior part to 4.3 Mpa in the posterior part.

A similar pattern of stress distribution was observed in the femoral head cartilage, albeit with a lower value of 2.9 Mpa. Notably, the von Mises stress exhibited a uniform distribution across the entire femoral head. The measured contact area in the acetabulum was 471.672 mm^2^.

In the dysplastic hip model, the von Mises stress distribution reached a value of 8.5 Mpa at the point of contact between the subluxated femoral head and the superior margin of the bony acetabulum. Similar to the normal hip model, the highest von Mises stress values were measured at the connection between the bony bridge and the iliac joint, as well as at the pubic bone, with a peak of 8.5 MPa. Additionally, stress concentration was observed in the Adams arch of the femoral neck. In the acetabulum, a von Mises stress concentration was found in a small area on the superior edge. The measured contact area in the dysplastic acetabulum was 205.272 mm^2^, which is less than half of the measured contact area in the normal acetabulum. On the femur, a similarly small area of stress concentration was located in the medial part of the femoral head, corresponding to the areas in contact with the acetabular edge.

The distributions of von Mises stress in each acetabulum model, without the femur, are depicted in Figure 4. The distributions of von Mises stress in each femoral head are illustrated in Figure 5.

To obtain a comprehensive overview of the position and relationship of the bones comprising the hip joint, the acetabulum/femur head volume ratio was calculated. The measurements clearly indicate higher values in the dysplastic hip, with ratios of 0.49 compared to 0.42 in the normal hip, as represented in Table 3.

Our findings indicate significantly elevated stress in the dysplastic hip model, which, in a real-life scenario, would likely result in premature articular cartilage wear and an earlier onset of arthrosis.

## 4. Discussion

Most finite element analyses (FEA) of hip joints have primarily focused on adult populations, examining pathologies such as femoroacetabular impingement, hip arthroplasties, and revision arthroplasties [26,27,28,29]. However, there is a noticeable dearth of FEA studies specifically centered on children with hip dysplasia. The limited research available in this area can be attributed to the technical and methodological challenges associated with creating accurate and reliable FEA models of pediatric hip joints.

The complex anatomy and dynamic growth patterns of pediatric hips pose unique challenges in accurately representing geometry and material properties in FEA models. Additionally, acquiring subject-specific data for children, including CT or MRI scans, is more complex compared to adults. These technical hurdles may have discouraged researchers from delving into this specific population.

Nonetheless, the lack of research underscores the importance of conducting studies in this domain to gain a better understanding of the biomechanics and treatment options for children with hip dysplasia.

The kidney-shaped contact surface located on the anterior–superior–posterior aspect of the femoral head and acetabulum is a well-recognized anatomical bony feature known as the “facies lunata”. In cases of dysplasia, we have observed a notably smaller, almost pinpoint-like, posterosuperior contact area. Various surgical procedures aim to enhance femoral head coverage and broaden the contact area. Alterations in acetabular inclination (such as Dega and Pemberton osteotomies), rotation (as in the Ganz method), and support area (like Salter and Chiari procedures) can be made. These adjustments can be finely tuned, and their impact can be assessed within the model, enabling the selection of the optimal correction method for each patient [30,31].

Despite these challenges, conducting FEA studies on normal and dysplastic hip joints in children is crucial. Hip dysplasia is a common condition in children and can lead to long-term complications if not properly managed. Understanding the mechanical behavior of the hip joint in children with dysplasia can provide valuable insights into the development and progression of the condition. FEA can also be used to evaluate the effects of various surgical interventions on the mechanical stress within the hip joint [22,29].

Understanding the mechanics and 3D geometry of normal and diseased hips is essential before embarking on a surgical procedure and subsequently devising a novel treatment approach. Evaluating the outcomes of surgical procedures, whether they are conventional or innovative, is a time-consuming process that raises ethical considerations. This is when 3D modeling and finite element analysis (FEA) methods can prove invaluable. We create and analyze a model of a healthy hip to comprehend its architecture and function under various real-life conditions. Subsequently, we endeavor to rectify the pathological model to approximate the normal one. The range of procedures available is virtually limitless because we can operate in a virtual environment free from the constraints of time and ethical considerations. Other authors have employed these methods to investigate various hip-related conditions, such as femoral head epiphyseolysis (SCFE), Perthes disease, and spastic hip [32].

For adult dysplastic hip joint analysis in 2010, Zhao et al. developed a patient-specific FEA model using MSC.Marc Mentat version 2005 R3 software [22]. They analyzed a single-leg stance and calculated the von Mises stress distribution in the acetabulum and femoral head. In their normal hip model, maximum von Mises stress values reached 13.06 MPa, whereas the dysplastic hip joint model showed a maximum measured von Mises stress of 19.90 MPa. This is consistent with our findings of increased cartilage stress, decreased contact area, and uneven stress distribution in the acetabular edge in dysplastic hips.

In the realm of single-leg stance analysis for dysplastic adult hip joints, Liu et al. recorded contact pressure values ranging from 5.5 to 18.9 MPa [33,34]. A similar study conducted by Zou et al. used ABAQUS version 2021 software for their FEA of dysplastic adult hip joints and found the lowest contact pressures to be in the range of 3.9 to 6.2 MPa [35].

To date, only a limited number of studies have employed finite element analysis of pediatric hip joints, and even fewer have focused on dysplastic pediatric hip joints. For instance, Park et al. constructed subject-specific FEA models of the pediatric pelvis in slipped capital femoral epiphysis and Legg–Calve–Perthes disease. In their normal hip model, the measured contact pressure was 1.87 MPa [36].

Kim et al. used CT and MRI data to reconstruct bony components and the cartilaginous surface in their FEA model for dysplastic pediatric hips. Their models employed linear elastic materials specific to children’s bony tissue, and the boundary conditions were set for a single-leg stance with vertical pelvic movement while the femur was fixed in all directions. In their analysis, the maximum contact pressure in the dysplastic hip joint was 6.0 MPa. In our dysplastic hip analysis, a higher maximum contact pressure of 8.3 MPa was observed, albeit with a smaller contact area [37].

In cases in which MRI data are unavailable, Anderson et al. demonstrated that the width of the articular cartilage can be approximated accurately by elevating the cortical bone of the articular components [14,38]. In our study, the cortical bone was elevated by 2 mm in each bony part, representing half of the distance between the acetabulum and the femoral head.

Skytte et al. employed a similar approach to model the pediatric hip of a 10-year-old girl. They used a CT scan of the slightly dysplastic right hip, whereas the left hip was considered normal. One of their models utilized the material properties of children’s bony tissue and maintained a similar cartilage height [39]. They also used material properties from the same source as Ohman et al. [16]. In their dysplastic model, the measured contact area closely resembled our measurements under a 300% bodyweight load, with peak contact pressures ranging from 11.0 to 14.7 MPa.

Although this study offers valuable insights, it has several limitations that need to be considered. Firstly, obtaining accurate and reliable subject-specific data for creating FEA models is challenging, especially for children. Pediatric hip joints undergo rapid growth and development, making it difficult to capture their complex anatomy. Additionally, acquiring imaging data, such as CT or MRI scans, in children can be more complex because of factors like sedation requirements and compliance issues. Moreover, the process incurs significant costs, encompassing expenses related to software and technician labor. Additionally, only a select few individuals possess expertise in both the intricate mathematical and biological aspects involved. The tasks of segmenting the slices, refining the geometric model, and analyzing the outcomes of various treatments are laborious and time-consuming, consuming precious time that patients do not have.

Another limitation pertains to the assumptions of material properties in FEA models. The mechanical properties of hip joint tissues, including bone, cartilage, and ligaments, can vary among individuals and are influenced by age, sex, and other factors. However, because of the lack of comprehensive data on pediatric hip joint material properties, simplifications and assumptions are often necessary in FEA studies, potentially introducing uncertainties and affecting result accuracy.

Moreover, FEA studies are based on mathematical models and simulations, which inherently carry limitations. Result accuracy depends on input parameter accuracy, boundary conditions, and assumptions made in the model. Small variations in these parameters can lead to significant differences in results. Therefore, it is important to validate FEA results with experimental data whenever possible to ensure their reliability.

Additionally, FEA studies may not fully capture the complex interactions between biological, physiological, and mechanical factors within the hip joint. The hip joint is a dynamic system influenced by factors such as muscle forces, joint kinematics, and patient-specific characteristics. FEA studies may not fully account for these dynamic interactions, limiting the comprehensive understanding of hip joint behavior.

Lastly, it is essential to recognize that this study includes a limited number of patients. The findings cannot be generalized to all patients, and this study serves as a foundation for further research involving larger, demographically diverse studies. Such studies have the potential to reshape our approach to managing severely impacted pathologies.

When examining the FEA results, we gain insight beyond mere geometric outcomes; we can assess how effectively our surgical procedures will function for the patient from a mechanical perspective. In particular, this study introduces a methodology for acquiring and contrasting 3D FEA models. Our primary objective was to generate results that correlate with real-world and the literature-based data. Future research endeavors will involve creating additional models for anthropometric studies, refining and “operating” on these models, and obtaining data for preoperative planning.

## 5. Conclusions

The use of finite element analysis in orthopedic surgery provides valuable insights into the mechanical behavior of anatomical structures. Subject-specific modelling offers distinct advantages, allowing surgeons to prepare for surgery tailored to the unique bone deformities that require correction. In cases of developmental hip dysplasia, joint morphology can vary significantly from one patient to another. Children with residual or untreated hip dysplasia often require complex pelvic or femoral osteotomies.

The geometric models, derived from patient-specific data, and the calculation of intraarticular contact areas and pressures through finite element analysis have enabled us to visualize problematic areas within the acetabulum or the proximal femur with remarkable clarity. Analyzing both normal and dysplastic hips in children using the finite element method has provided us with valuable insights for quantifying the pathology of dysplastic hips. This knowledge can be instrumental in the development of osteotomies and allows for the quantification and comparison of the effectiveness of various surgical treatments employed in the management of hip dysplasia in children.

## Figures and Tables

**Figure 1 jpm-13-01593-f001:**
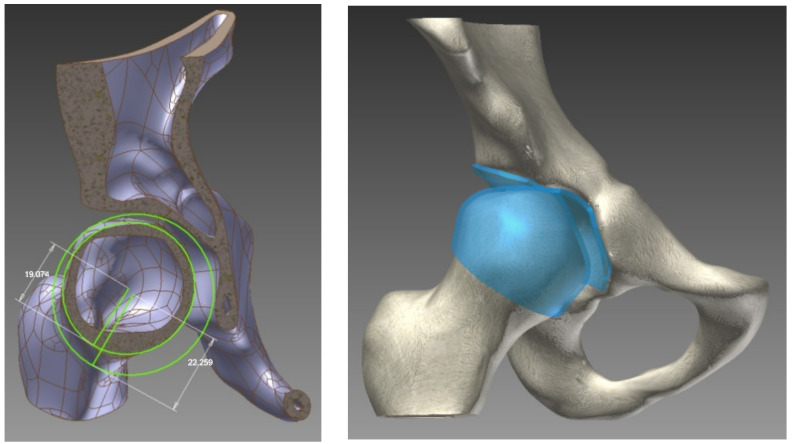
The geometrical model: the creation and position of the joint cartilage.

**Figure 2 jpm-13-01593-f002:**
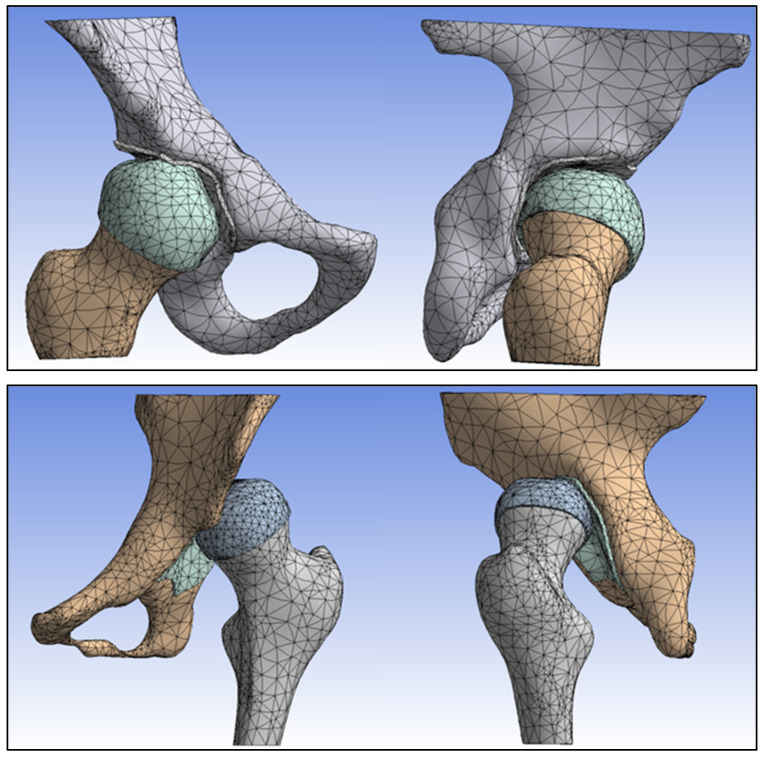
The numbers of hexagon elements and nodes in the normal and dysplastic hips.

**Figure 3 jpm-13-01593-f003:**
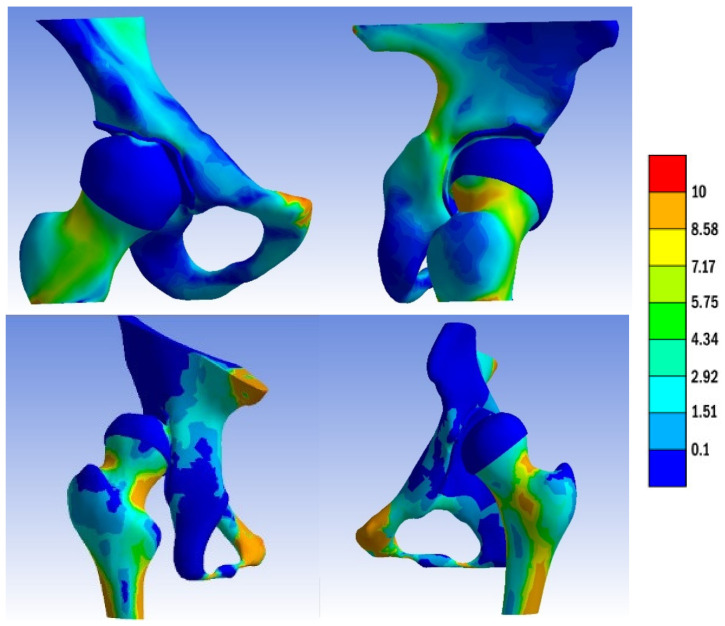
The von Mises stress distribution in the normal and dysplastic hip.

**Figure 4 jpm-13-01593-f004:**
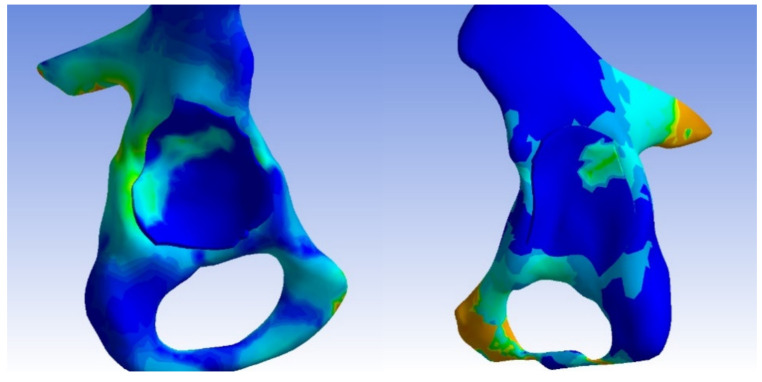
The distribution of the von Mises stress in normal and dysplastic acetabula.

**Figure 5 jpm-13-01593-f005:**
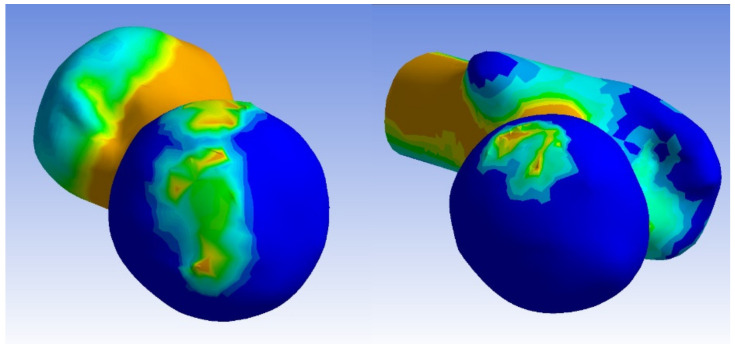
The distributions of von Mises stress in the normal and dysplastic femoral heads.

**Table 1 jpm-13-01593-t001:** Children’s bone tissue material properties used in the mathematical models.

	Density (g/cm^3^)	Young	Poisson	Yield Tensile Strength (Mpa)	Yield Compressive Strength (Mpa)	Ultimate Tensile Strength (Mpa)	Ultimate Compressive Strength (Mpa)
**Cortical bone**	1.05	11.22 Gpa	0.3	35	62	114	135
**Trabecular bone**	0.60	5.33 Mpa	0.2	7	15	30	80
**Cartilage**	1.1	0.6 Mpa	0.45	1	1.5	4.4	20

**Table 2 jpm-13-01593-t002:** The numbers of hexagon elements and nodes used for finite element analysis.

No. Nodes/Elements	Cortical Bone Pelvis	Trabecular Bone Pelvis	Cartilage Pelvis	Cortical Bone Femur	Trabecular Bone Femur	Cartilage Femur	Mesh
Normal	20,147/11,233	9787/5007	2104/1000	10,528/5710	6440/3596	4576/2298	55,627/29,907
Dysplastic	16,635/8978	8328/4500	1695/802	11,630/6341	5852/3255	5031/2500	51,436/27,594

**Table 3 jpm-13-01593-t003:** Acetabulum/femoral head volume ratio.

	Acetabulum (mm^3^)	Femoral Head (mm^3^)	Acetabulum/Femoral Head Volume Ratio
**Normal**	12,095.928	28,360.439	0.42
**Dysplastic**	8393.275	22,325	0.49

## Data Availability

Data are contained within the article.

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
