# Peer review of "Finite Element Analysis of Normal and Dysplastic Hip Joints in Children"

_jpm, 2023, doi:10.3390/jpm13111593_

Round 1

Reviewer 1 Report

Comments and Suggestions for Authors

1."What are the key applications of finite element analysis (FEA) in the study of hip joints in children, and how does it contribute to improving the treatment of hip joint conditions?

2. "What are the key differences in mechanical properties between child and adult bones, and how do these differences impact the finite element analysis of hip joint models in children?"

3."What are the key stress distribution differences between normal and dysplastic hip models, and how do they relate to hip dysplasia's mechanical effects?"

  1. 4. "How do the observed stress patterns in the acetabulum and femoral head cartilage inform our understanding of hip joint mechanics, especially in dysplasia, and their relevance for treatment strategies?"

  2.  
  3. 5. "What are the challenges and limitations associated with FEA studies on pediatric hip joints, and how do they impact the accuracy of results for hip dysplasia treatment?"
    1. 6. "In comparing FEA studies on dysplastic adult and pediatric hip joints, what are the key findings, and how do they shape our understanding of hip dysplasia biomechanics and potential treatments for children?"

    2.  
    3. 7. "What are the limitations of FEA studies in pediatric hip joint research, and how do these limitations affect result reliability?"
      1. 8."How well do FEA studies address the dynamic complexities of the hip joint, and what are the implications for our understanding and potential treatment approaches, considering the small sample size in this study?"

Author Response

Dear reviewer,

We would like to thank you and the reviewers for your insightful comments, which have greatly helped us to improve the quality of our manuscript.

We would like to address the comments and suggestions point by point:

Point 1. We have added a paragraph addressing this point in line 325-335. 
Point 2. A paragraph has been added to the revised version of the manuscript addressing this issue in line 139-148. 
Point 3. Thank you for your comment. A paragraph was added in line 290-292 addressing this issue. 

Point 4. We are grateful for your comment. The discussion section was extended with a paragraph in line 310-318 in this concern. 

Point 5. The limitations-paragraph of our study were further enriched with and addition of line 382-387, concerning your insightful comment.

Point 6. We appreciate the point made, and made accordingly a change in line 102-107. 

Point 7. We appreciate the comment, but we feel that the addition to the limitations in line 382-387 also cover this point. 

Point 8. We are grateful for your comment, and added another paragraph in the discussion section that covers this point, in line 408-414. 

Thank you for all the insightful comments and points made. We made changes accordingly and hope to bring the manuscript a step closer to a publication version, thanks to your review. 

Kind regards, 
Vlad Vunvulea

Reviewer 2 Report

Comments and Suggestions for Authors

Dear Authors:

I want to congratulate with You for Your work. Some revision is needed, in my opinion, before the paper can be accepted for publication, but they are relative to English language. The only other point is: consent and reason to CT scan acquisition in the two children must be specified.

Comments on the Quality of English Language

Passive form should be used instead of "we"

Author Response

Dear reviewer,

Thank you very much for taking the time to review this manuscript.

We would like to thank you and the reviewers for your insightful comments, which have greatly helped us to improve the quality of our manuscript.

We are thankful for pointing out the miss-use of the first plural voice. We made changes accordingly, and used passive voice in the manuscript. 
Thank you as well for the second point, we have added the reason and consent for the CT scan acquisition in line 129-131.

Thank you for the kind comments. We are grateful for your points, and we hope that the changes made bring the manuscript to a closer format to publication. 

Kind regards, 
Vlad Vunvulea